**Subject Category:**
Biology (whole organism)

evolution/ecology/genetics

Atlantic salmon, domestication, age, kype, sexual selection, allometry

**Author for correspondence:**
William Bernard Perry
e-mail: w.perry@bangor.ac.uk

# Evolutionary drivers of kype size in Atlantic salmon (*Salmo salar*): domestication, age and genetics

William Bernard Perry[1], Monica Favnebøe Solberg[2], Francois Besnier[2], Lise Dyrhovden[3], Ivar Helge Matre[3], Per Gunnar Fjelldal[2], Fernando Ayllon[2], Simon Creer[1], Martin Llewellyn[4], Martin I. Taylor[5], Gary Carvalho[1] and Kevin Alan Glover[2,6]

[1]Molecular Ecology and Fisheries Genetics Laboratory, School of Natural Sciences, Bangor University, Bangor, Gwynedd LL57 2UW, UK
[2]Population Genetics Research Group, Institute of Marine Research, P.O. Box 1870, Nordnes, NO-5817 Bergen, Norway
[3]Matre Research Station, Institute of Marine Research, Matredal, Norway
[4]Institute of Biodiversity, Animal Health and Comparative Medicine, University of Glasgow, Glasgow G12 8QQ, UK
[5]School of Biological Sciences, University of East Anglia, Norwich NR4 7TJ, UK
[6]Institute of Biology, University of Bergen, N-5020 Bergen, Norway

(iD) WBP, 0000-0001-9596-3333; MIT, 0000-0002-3858-0712; GC, 0000-0002-9509-7284

The diversity of reproduction and associated mating patterns in Atlantic salmon (*Salmo salar*) has long captivated evolutionary biologists. *Salmo salar* exhibit strategies involving migration, bold mating behaviours and radical morphological and physiological change. One such radical change is the elongation and curvature of the lower jaw in sexually mature males into a hook-like appendage called the kype. The kype is a secondary sexual characteristic used in mating hierarchies and a prime candidate for sexual selection. As one of the core global aquaculture fish species, however, mate choice, and thus sexual selection, has been replaced by industrial artificial fertilization seeking to develop more commercially viable strains. Removal of mate choice provides a unique opportunity to examine the kype over successive generations in the absence of sexual selection. Here we use a large-scale common-garden experiment, incorporating six experimental strains (wild, farmed and wild × farmed hybrids), experiencing one to three sea winters, to assess the impact of age and genetic background.

After controlling for allometry, fork length-adjusted kype height (AKH) was significantly reduced in the domesticated strain in comparison to two wild strains. Furthermore, genetic variation at a locus on linkage group SSA1 was associated with kype height, and a locus on linkage group SSA23 was associated with fork length-adjusted kype length (AKL). The reduction in fork length-AKH in domesticated salmon suggests that the kype is of importance in mate choice and that it has decreased due to relaxation of sexual selection. Fork length-AKL showed an increase in domesticated individuals, highlighting that it may not be an important cue in mate choice. These results give us insight into the evolutionary significance of the kype, as well as implications of genetic induced phenotypic change caused by domesticated individuals escaping into the natural environment.

## 1. Introduction

Atlantic salmon (*Salmo salar*) are predominantly anadromous salmonid fish that inhabit coldwater streams on both sides of the northern Atlantic during the freshwater stage of their life cycle. The species is known to display phenotypic variation among individuals and populations, some of which may be adaptive [1]. During its life cycle, it undergoes key phenotypic changes manifested in variable morphology [2], physiology [3] and behaviour [4]. Such fundamental biological change is required for survival in both freshwater and marine environments; environments with radically different abiotic and biotic conditions, ranging from differences in salinity and temperature profiles, to changes in competition with conspecifics. The wealth of life-history variation, both among and within populations [5], renders Atlantic salmon well suited for investigating the roles of phenotypic plasticity and heritable genetic change in generating variation for maximized reproductive success [6].

A critical stage in the life cycle of Atlantic salmon, closely linked with fitness, is spawning. After returning to natal freshwater streams and rivers during the summer and autumn months, females will excavate depressions in the river bed to form nests, where they lay eggs which are simultaneously fertilized by the milt of one or more males [7]. Males are able to spawn numerous times in quick succession for up to two months [8], unlike females who spawn over a more limited time period, resulting in male-biased operational sex ratios [9]. The disparity between mating males and fertile females generates intense male–male competition, fuelled by the increased fertilization success seen in those males who fertilize eggs first [10].

A phenotypic trait that has been associated with the intense male competition during spawning is the kype, an elongation of the lower jaw forming a hook at the tip (figure 1*b*). Darwin used the male kype in salmon as evidence for the role of sexual traits in natural selection, referencing its defensive use during altercations among spawning males [11]. A century later, Jones [7] refers again to the kype as a weapon for defence, likening the structure to the antlers of a stag, adding that bodily harm is rarely seen, with most conspecific altercations resolved with agonistic displays. Since these early descriptions, some novel discoveries surrounding the kype have been made [2,12], though the fundamental understanding of its purpose has remained mostly true to the definition put forward by Darwin. Behavioural experiments in salmonids have revealed some evidence that the kype is used in both intra- and inter-sexual interactions, with correlations seen between kype length and (i) increased rank within local male dominance hierarchies and (ii) increased female mate choice [9,12]. However, due to issues of correlation with other key characteristics associated with female mate choice, such as body size, these results are not only inconclusive but are difficult to disentangle in behavioural experiments. Other experimental approaches are required to contribute to our understanding of the significance of the kype in Atlantic salmon breeding systems, as well as its evolution.

Since the early 1970s, and for more than 12 generations, Atlantic salmon have been subject to domestication and directional selection for economically important traits. As a result, domesticated salmon now display a contrasting array of genetic and phenotypic differences to wild Atlantic salmon [13]. Salmon breeding programmes operate by the selection of individuals according to their breeding values, manual stripping of gametes, and thereafter controlled fertilization for production of families for the next generation of selection. In effect, this practice removes all opportunity for spawning competition and mate choice, and thus renders the development of secondary sexual characteristics involved in mate choice or sexual success, potentially redundant [14–16]. Therefore, studying domesticated and wild salmon under controlled conditions may yield insights into the evolutionary

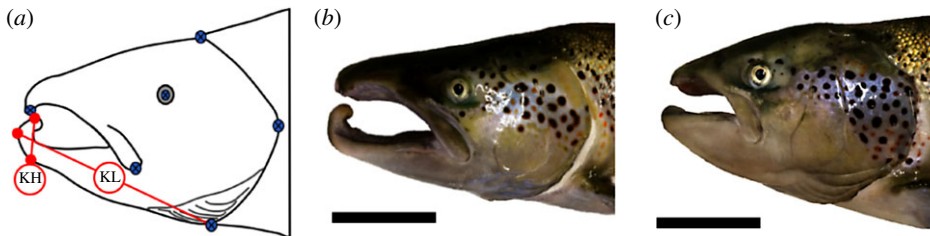

**Figure 1.** (*a*) Landmarks (blue crosshair) used for the geometric morphometric analysis, kype length (KL) and kype height (KH); in addition to examples demonstrating variation in head morphology, including a mature male showing an elongated kype (*b*) and a mature male with a reduced kype (*c*) (both mature males are 2SW). Scale bars represent 10 cm.

significance of head morphology and the kype in Atlantic salmon, as well as wider impacts of domestication.

The few studies examining the kype of male Atlantic salmon and its role in reproduction have focused on behaviour in wild individuals [12,17]. While studies in the wild are informative, opportunities that examine the impact of exposure to vastly different selection regimes in captivity offer an experimental framework for environmental and genetic manipulation under controlled conditions. Such empirical data are of additional interest in the context of domesticated escapees, that typically display a lower spawning success than wild salmon [18,19]. It remains unclear, however, the degree to which the lower domesticated male spawning success results from phenotypic limitations arising from shifts in head morphology and kype characteristics.

Common-garden experiments, whereby individuals of the differing genetic background are reared under identical conditions, can elucidate the degree of genetic influence on traits of interest. Such studies are common in Atlantic salmon, revealing, among other things, differences between domesticated and wild salmon in traits of evolutionary significance such as growth [20,21], survival in the wild [22,23] and precocious male maturation [24]. Here, we use a pedigree-controlled multi-generation population of domesticated, hybrid and wild Atlantic salmon, alongside a quantitative trait loci (QTL) analysis, to explore shifts in sexually selected traits influenced by domestication. We implemented both classical and geometric morphometrics on 528 sexually mature adult male Atlantic salmon, of varying age (experiencing one, two and three sea winters (SWs)), to investigate whether domestication, and its associated relaxation of sexual selection, has led to detectable changes in head morphology in approximately 12 generations.

# 2. Material and methods

## 2.1. Fish

A range of common-garden experiments on domesticated, hybrid and wild Atlantic salmon have been conducted at the Institute of Marine Research (IMR) for more than a decade (e.g. [20,25–27]). In this process, a large pedigree-based population of Atlantic salmon consisting of fish originating from multiple wild populations and domesticated strains, including respective hybrids and backcrosses, has been established. Individuals are reared under standard farming conditions (i.e. tanks in fresh water, sea cages for the marine stage), where strains are mixed from the eyed-egg stage onwards. Fish are then allowed to mature naturally under a natural day-length for Bergen at age 1, 2 or 3+ SWs, all of which smoltified at age 1 (thus total age = SW age + 1).

In the present study, we used sexually mature adult males, of which the genetic sex of all individuals was validated by a DNA probe-based RTPCR presence/absence assay (Thermo Fisher Scientific, USA) aimed to detect the presence of the male-specific *sdY* gene [28,29]. The fish originated from the first generation of the main domesticated–wild population established at IMR in 2011 and includes fish from three wild populations, a domesticated strain and a domesticated–wild F1 hybrid strain (table 1). Artificial fertilization of gametes took place on arrival at the IMR Matre Research station. For further details regarding the parental lines, rivers of origin and production of these fish, see Solberg *et al*. [30] (information on the cohort produced in 2012). Fish were reared under the experimental protocol (ID 5296) that was approved by the Norwegian Animal Research Authority (NARA). Procedures included DNA identification and subsequent PIT tagging of all individuals for identification. Upon termination,

**Table 1.** Background of experimental fish, including strains, geographic origin of wild strains, number of families comprising a strain and number of individuals. Numbers inside brackets represent individuals used in kype height analysis, numbers outside the brackets represent individuals used in kype length analysis. Males: established on 16 and 22–23 November 2011. Hatched in Spring 2012. Individuals maturing as 1SW, 2SW and 3SW were terminated on 27 January 2015, 18 January 2016 and 17 January 2017, respectively. Females: siblings to males. Terminated when ripe (November–January 2016).

| sex | type | origin | 1SW families (n) | 1SW individuals (n) | 2SW families (n) | 2SW individuals (n) | 3SW families (n) | 3SW individuals (n) |
|---|---|---|---|---|---|---|---|---|
| male | farmed | Mowi | 5 (4) | 14 (12) | 4 (4) | 16 (12) | 4 (4) | 14 (14) |
| | hybrid | Figgjo (♀)×Mowi (♂) | 6 (6) | 95 (91) | 5 (4) | 8 (7) | 5 (4) | 13 (11) |
| | | Mowi (♀)×Figgjo (♂) | 7 (7) | 91 (85) | 6 (4) | 18 (11) | 4 (4) | 4 (4) |
| | wild | Arna 60°42′ N, 5°46′E | 6 (6) | 63 (62) | 4 (4) | 11 (10) | 3 (3) | 7 (6) |
| | | Figgjo 58°81′ N, 5°55′ E | 6 (6) | 64 (61) | 6 (5) | 8 (6) | 0 | 0 |
| | | Vosso 60°64′ N, 5°95′ E | 7 (7) | 70 (67) | 6 (6) | 20 (12) | 5 (4) | 12 (9) |
| | total | | 37 (36) | 397 (378) | 31 (27) | 81 (58) | 21 (19) | 50 (44) |
| female | farmed | Mowi | | | 0 | 0 | | |
| | hybrid | Mowi (♀)×Figgjo (♂) | | | 7 | 32 | | |
| | | Figgjo (♀)×Mowi (♂) | | | 5 | 14 | | |
| | wild | Arna | | | 3 | 13 | | |
| | | Figgjo | | | 4 | 11 | | |
| | | Vosso | | | 2 | 4 | | |
| | | unknown | | | — | 3 | | |
| | total | | | | 21 | 77 | | |

fish were sedated using Aqui-S, then killed using an anaesthetic overdose of MS-222 and bled by cutting one of the gills. Those working directly with the experimental animals had also undergone Norwegian Food Safety Authority (NFSA) training, as is required with experimentation involving animals that are included in the Norwegian Animal Welfare Act (2010).

## 2.2. Data collection

Photographs of the lateral side of 1SW, 2SW and 3SW mature male salmon were taken at the end of the spawning season (i.e. January) in 2015, 2016 and 2017 (table 1) using a mounted digital single lens reflex camera and measurement board. Mature female fish, used as an outlier group for the geometric morphometrics analyses, were terminated and photographed when they were ripe through the course of the spawning season in the winter of 2014/2015, thus having completed two SWs. Milt weight, fork length (most anterior point of the head to the of the middle caudal fin rays) and total wet weight of the mature adult males were also taken during sampling. Milt weight and total wet weight were log10 transformed and used in a linear regression to calculate gonadosomatic residuals (GSR). Internal PIT tags were scanned, which allowed for the pedigree of the fish to be unequivocally identified. Photographs were filtered by technical quality before the application of landmarks and before the individual fish were identified, allowing for the unbiased removal of images. After filtering, the final mature male dataset used for both kype length and geometric morphometrics included 397 1SW, 81 2SW and 50 3SW males from three wild populations, a domesticated strain and a reciprocal F1 hybrid population (a total of 37 families represented in the pedigree) (table 1). The mature male photographs were also supplemented by 77 2SW mature females for use in the geometric morphometric analysis, represented by 21 families from the same origins as the males. After the removal of photographs in which the kype was obscured (closed mouths), a subset of individuals was used for kype height comparisons, comprising 378 1SW, 58 2SW and 44 3SW males. All subsequent analysis of photographs was undertaken without prior knowledge of the genetic background of fish. Additionally, the sequence in which photographs were analysed within SWs was ordered randomly.

## 2.3. Geometric morphometrics

Positioning of landmarks for the geometric morphometric analysis was based on identifiable external features related to the skeletal form of the head [31,32], many of which have been outlined in the previous literature documenting both salmonid and other fish morphology. These landmarks ($n = 6$) included the apex of the upper jaw, the most dorsal [33] and ventral positions of the gill plate, the most posterior point of the gill plate, the maxillary bone [33] and the eye (figure 1a). Although of interest, landmarks positioned on the kype were not used in the geometric morphometric analysis due to the influence of mouth opening on head shape. Landmarks were applied by the same observer, using tpsDig v. 2.28 [34].

Landmark data were analysed using the R package geomorph 2.0 [35]. Generalized Procrustes analysis (GPA) was conducted on landmark data to limit the effect of scale, orientation and translation between images. Aligned Procrustes coordinates for all landmarks in all individuals were then verified by plotting points around the mean value (electronic supplementary material, figure S1). A principal component analysis (PCA) was then conducted on the transformed coordinates for 1SW, 2SW and 3SW individuals, together with 95% confidence ellipses around respective SWs, assuming a multivariate $t$-distribution. All statistical analyses were carried out in R v. 3.3.3 [36] and are presented in electronic supplementary material, file S1.

## 2.4. Linear measurements

Linear length measurement of the lower jaw (referred to here as kype length), as well as length of the hook forming at the tip of the lower jaw (referred to here as kype height), were taken—both of which comprise the characteristic kype (figure 1a). Length of the lower jaw was taken from the most anterior position of the lower jaw to the bottom of the gill plate. Kype height was taken from the most dorsal peak of the hook to the ventral position on the lower jaw where curvature began, as used in previous studies [12,15,37]. If no clear curvature was found on the lower jaw, the point directly beneath the dorsal peak was used. Lengths were calculated from landmarks placed at these positions.

Single linear regressions were conducted to assess the relationship between fork length and kype length, as well between kype height and fork length, all of which were log10 transformed.

The residuals from the regression were used to produce a fork length-adjusted kype length (AKL) and a fork length-adjusted kype height (AKH).

To assess factors influencing AKL and AKH, two mixed effect models (LME) were constructed using the R package 'lme4' [38], one for each response variable. The full models contained the fixed factors: SW, strain and GSR, as well as their two-way interactions, with the random intercept factors: family (nested within strain type, to control for the hierarchical structure of the data), sire and dam. The 'step' function within 'lme4' was then used to select the best fitting model through automatic backward elimination, allowing for the removal of fixed terms and random factors which did not contribute to the model.

Analysis of variance type III sum of squares with Satterthwaite approximation for degrees of freedom allowed for the generation of p-values between factors in mixed effect models, using the package 'lmerTest' [39]. Sire was added as a random effect to the best fitting model for AKL to calculate with Satterthwaite approximation for degrees of freedom using 'lmerTest' but was removed as a random effect for all other aspects of the analysis. Estimated marginal means and pairwise comparisons between means were calculated using the selected models and the R package 'emmeans' [40]. A Tukey's multiple comparisons test was used to adjust p-values, and Kenward–Roger approximations were used to estimate degrees of freedom.

## 2.5. Quantitative trait loci analysis

All individuals with phenotype measurements, and their parents, were genotyped with 109 SNP markers evenly distributed on the 29 chromosomes of the salmon genome [41]. The identity by descent (IBD) relation between offspring was estimated by using both the genotype and pedigree information [42]. A hierarchical generalized linear model (HGLM) [43] was then fitted, at each locus, to test for correlation between kype measurement and genotype. $y = XB + Ga + e$ (model 0), $y = XB + Ga + Zq + e$ (model 1), where $X$ is the model matrix for fixed effects (tank, strain and SW), $B$ the vector of fixed effect, $G$ the kinship matrix, $a$ the vector of polygenic effects, $Z$ the locus-specific IBD matrix, $q$ the vector of QTL effect and $e$ the residuals. To test for a genotype–phenotype correlation at each locus, the likelihood of the model without QTL effect (model 0) and with QTL effect (model 1) were compared in a likelihood ratio test. Each model was fitted in R using the HGLM package [43].

# 3. Results

## 3.1. Overview

Kype length, kype height (figure 1a) and whole-fish fork length were collected from a total of 528 males originating from six experimental strains that had been reared together from the eyed-egg stage until maturity at 1SW–3SW. To control for allometry, kype length and kype height were adjusted for fork length (after being log10 transformed), and the residuals generated from an ordinary least-squares linear regression between the two variables were used to produce fork length-AKL and fork length-AKH. Sexing of the fish revealed that three individuals used in the study were genetically female, despite these individuals producing a considerable weight of milt (milt weight range: 50.9–232.4 g). The three individuals belonged to three different strains (domesticated, wild and hybrid) and were kept in the analysis after visually assessing them for female phenotypic traits.

## 3.2. Kype length and height

The observed raw kype length and height both showed marked variation within the whole dataset containing all SWs and all strains, ranging between 5.2 and 28.1 cm (raw kype length) and 0.2 and 6.1 cm (raw kype height). Variation in kype height and kype length was also evident from looking at individual photographs, even within SWs (figure 1b,c) and has been documented in previous studies [44]. From raw data, it was also clear that variation existed among strains, both in terms of kype length and height (figure 2). Kype length and height were, however, highly correlated with fork length ($R^2 = 0.86$, $F_{1,526} = 3284$, $p < 0.01$; $R^2 = 0.78$, $F_{1,478} = 1680$, $p < 0.01$, respectively) (figure 3), a correlation that was even stronger when values were log10 transformed ($R^2 = 0.87$, $F_{1,526} = 3490$, $p < 0.01$; $R^2 = 0.81$, $F_{1,478} = 2036$, $p < 0.01$, respectively). Residuals from the log-transformed regressions were used as a fork length-AKL and fork length-AKH.

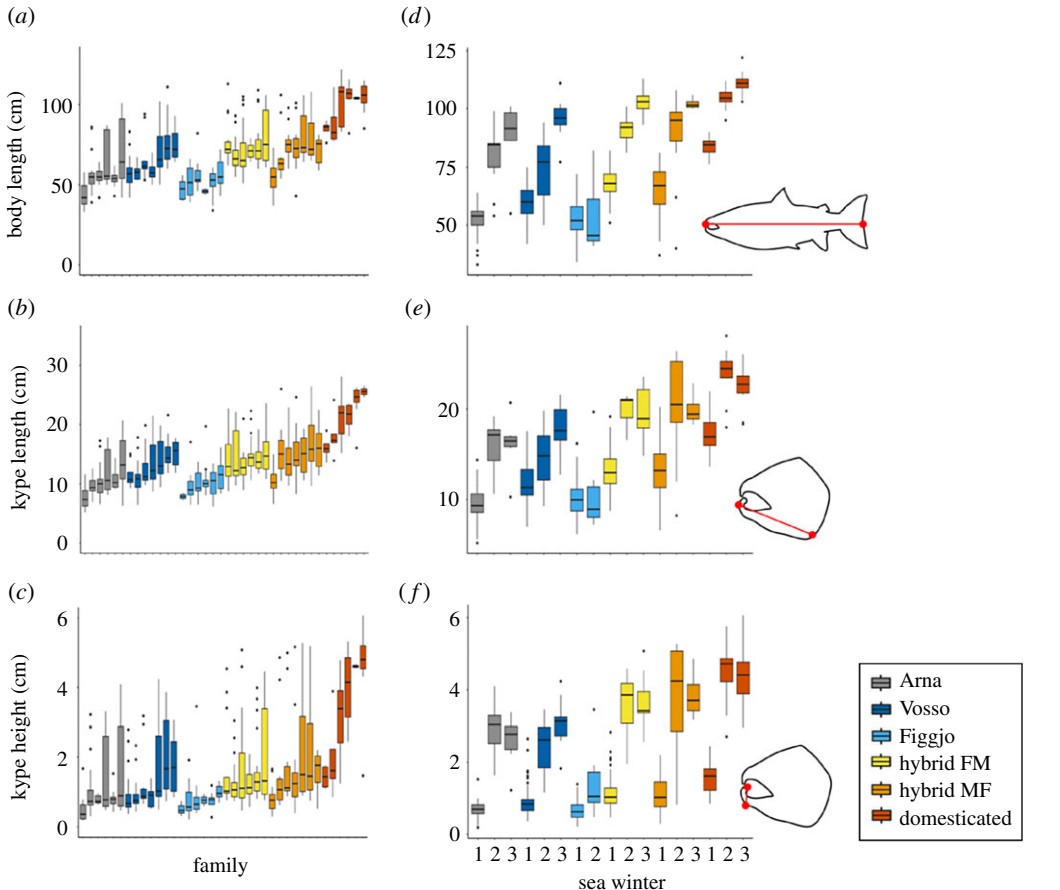

**Figure 2.** Boxplot of the observed variation in fork length, kype length and kype height broken down by family ($a-c$) and SW ($d-f$). Strains consist of wild (Arna, Vosso and Figgjo), hybrid (hybrid Figgjo × Mowi and hybrid Mowi × Figgjo) and domesticated genetic backgrounds.

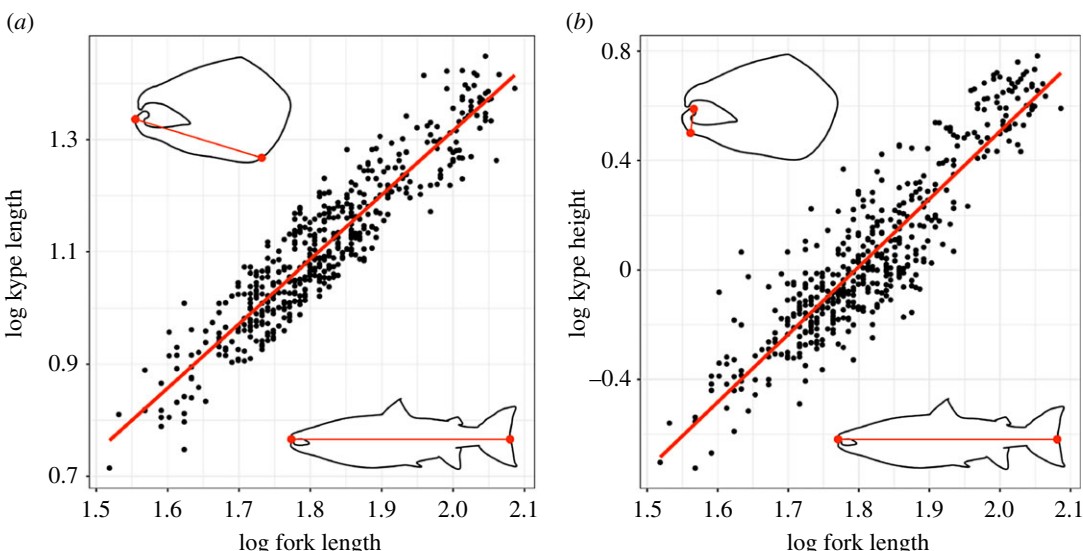

**Figure 3.** ($a,b$) Linear regression between log10 fork length and log10 kype length/height, including individuals from all SWs and strains. Residuals from these regressions were used for fork length-AKL and fork length-AKH.

For AKL, the selected model contained terms: strain and SW. AKL showed significant differences among strains (LME strain: $F_{5,52} = 4.37$, Sum Sq = 0.050, $p < 0.01$) (figure 4b). The difference in AKL among strains was driven by the significantly larger AKL in the domesticated strain (estimated

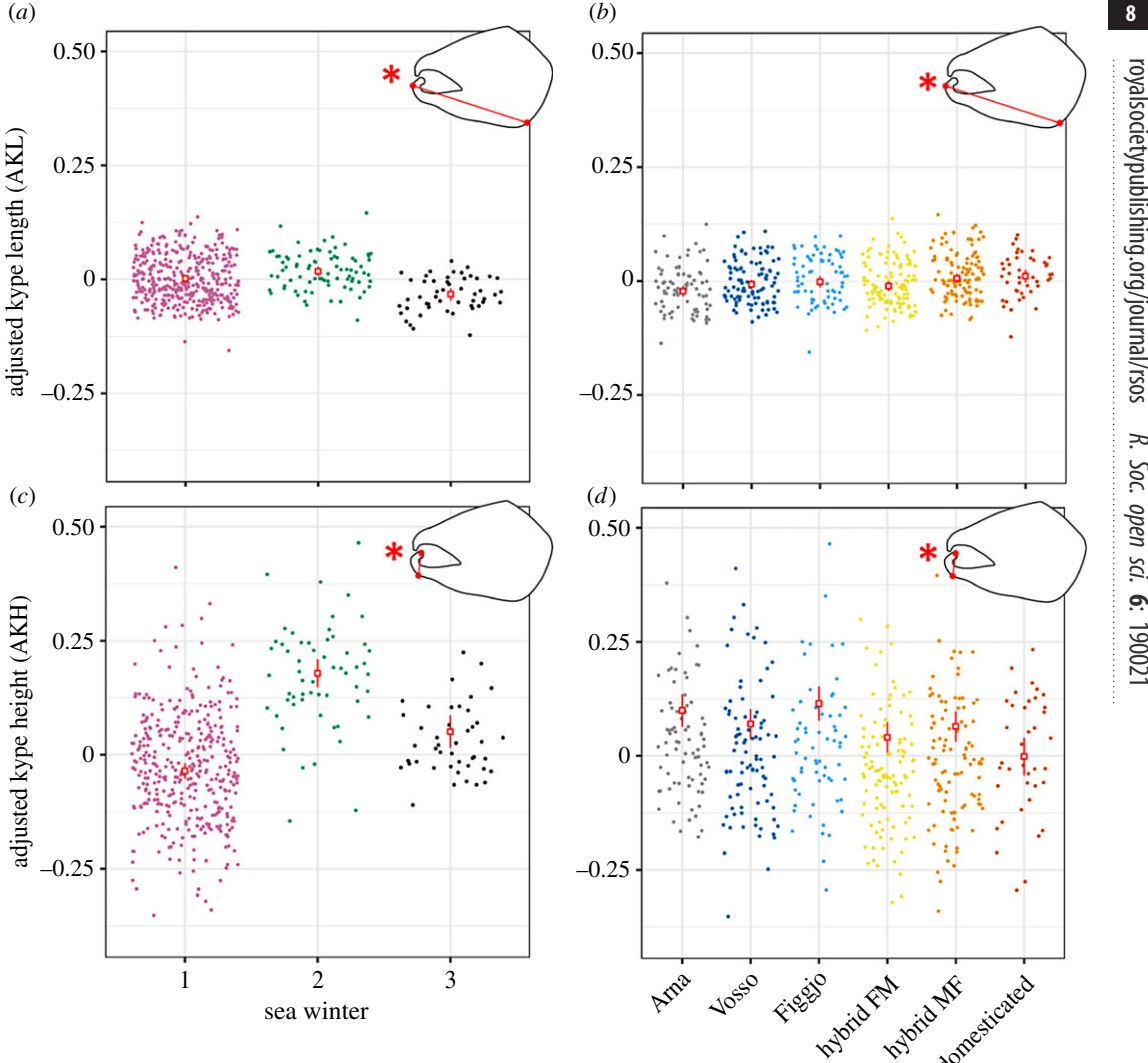

**Figure 4.** Fork length-AKL (*a,b*) and fork length-AKH (*c,d*) broken down by SW and strain. Red asterisks represent a significant effect ($p \leq 0.01$) of the factor displayed on the x-axis for AKL or AKH, as shown from the linear mixed effect model.

mean $= 0.012$) when compared to Arna (estimated mean $= -0.022$, $t_{63} = 3.29$, $p = 0.02$), as well as the significantly larger AKL seen in the hybrid MF strain (Mowi ♀ × Figgjo ♂) (estimated mean $= 0.0057$, $t_{30} = 3.30$, $p = 0.03$), when compared to Arna. No other significant effects ($p < 0.05$) in AKL were found within the pairwise comparisons among strains (electronic supplementary material, table S1). AKL also varied among SWs (LME SW: $F_{2,518} = 17.16$, Sum Sq $= 0.08$, $p < 0.01$) (figure 4*a*), with AKL peaking at 2SW (estimated mean $= 0.018$), after displaying an increase from 1SW (estimated mean $= 0.002$, $t_{520} = 2.66$, $p = 0.02$). A large significant decrease in AKL was also detected between 2SW and 3SW (estimated mean $= -0.032$, $t_{519} = 5.82$, $p < 0.01$).

For AKH, the selected model contained the fixed effect terms strain and SW, as well as sire as a random factor. Strain had a significant effect on AKH (LME strain: $F_{5,49} = 4.78$, Sum Sq $= 0.282$, $p < 0.01$) with the domesticated strain (estimated mean $= -0.001$) showing a decrease in mean AKH when compared to the wild strains Arna (estimated mean $= 0.100$, $t_{60} = 3.74$, $p < 0.01$) and Figgjo (estimated mean $= 0.115$, $t_{76} = 4.16$, $p < 0.01$) (figure 4*d*). A reduction in AKH between the domesticated strain and the wild Vosso strain was also detected, although this did not show strong significance (mean $= 0.070$, $t_{62} = 2.71$, $p = 0.09$). Finally, a significant reduction in AKH was also seen between Figgjo and the hybrid FM strain (Figgjo ♀ × Mowi ♂) (estimated mean $= 0.040$, $t_{32} = 4.16$, $p < 0.01$). No other significant effects ($p < 0.05$) were found within the pairwise comparisons among strains (electronic supplementary material, table S1). A significant effect of SW on AKH was observed

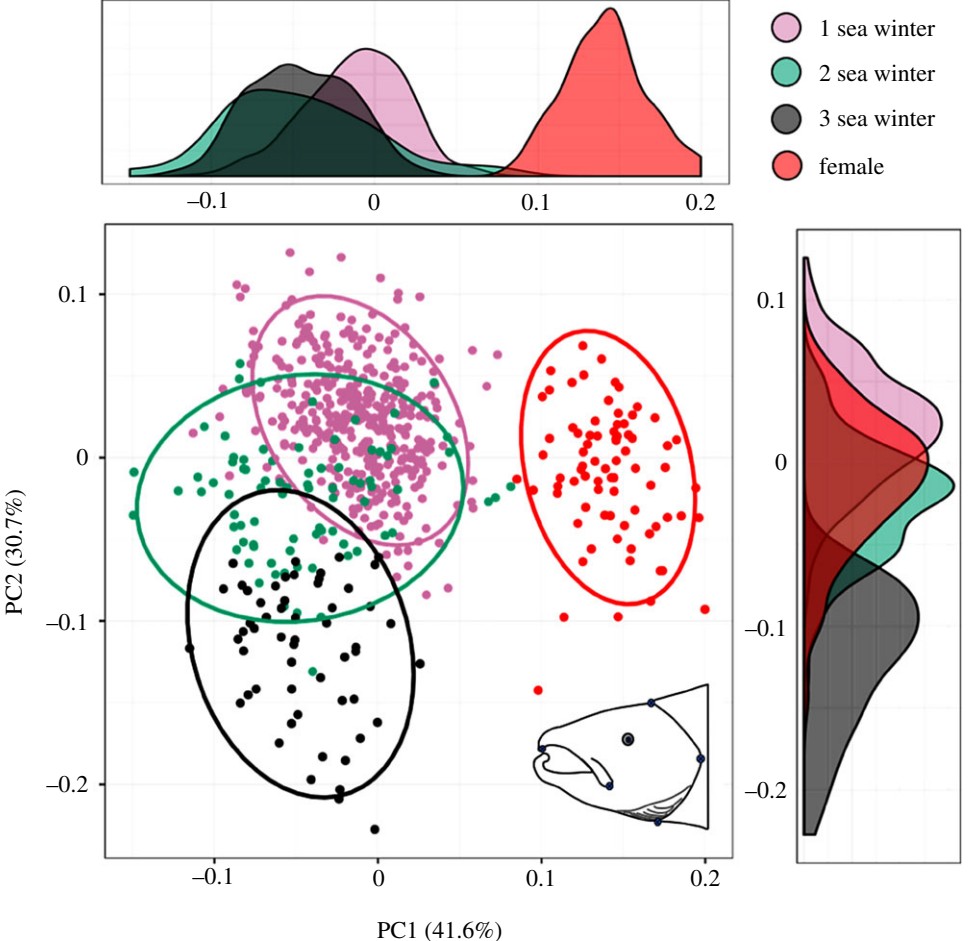

**Figure 5.** Principal component plot summarizing the greatest variance in morphospace of salmon head morphology, as represented by the six landmarks outlined in figure 1a. Groups are split into 1SW–3SW mature males, and 2SW mature females. Groups are ellipsed by 95% confidence intervals. Principal coordinate density plots have also been included on the x- and y-axes to better illustrate distribution between groups.

(LME strain: $F_{2,469} = 91.85$, Sum Sq = 2.17, $p < 0.01$), with an increase seen between 1SW (estimated mean = $-0.036$) and 2SW (estimated mean = 0.179) ($t_{470} = 13.36$, $p < 0.01$), with 2SW having the highest AKH estimated mean, with a decrease in 3SW (estimated mean = 0.051) when compared to 2SW ($t_{467} = 5.76$, $p < 0.01$) (figure 4c).

## 3.3. Geometric morphometrics

There was visible variation in male head shape (figure 1b,c) with some individuals showing shorter kype height, a feature shared with female head morphology. The results shown by the geometric morphometrics, however, showed no overlap between female and male head shape (figure 5). The lack of overlap in head shape between these groups indicates that while some males exhibit a reduced kype height, this morphological characteristic alone does not constitute female head morphology.

Influence of SW on head shape was seen in the landmark-based PCA plot, where a predominant difference in clustering can be seen between individuals belonging to 1SW/2SW and 3SW (figure 5). Differences in head shape are smaller between individuals from 1SW and 2SW, with distribution ellipses showing a larger proportion of overlap. Head shape was also assessed using the same methods among strains; however, no clustering was identified (electronic supplementary material, figure S2). The differences in male head shape between SWs were summarized along PC1 (proportion of variance = 46.5%) and were characterized by a dorsal shift of features such as the eye, posterior point of the gill plate and the maxillary bone (electronic supplementary material, figure S3).

## 3.4. Quantitative trait loci analysis

Scanning the genome for regions correlated with kype length did not return any significant QTL, though the scan for AKL revealed one QTL on linkage group SSA23 (electronic supplementary material, figure S4a). This QTL is for a large part caused by a strong correlation between AKL and the SNP haplotype of each offspring from parent F8. The AKL value of offspring inheriting haplotype 1 and haplotype 2 from F8 was (mean ± s.d.) −0.03 ± 0.05 and 0.02 ± 0.04, respectively ($t = −3–3$, d.f. = 37 $p = 0.002$). Estimated relative proportion of QTL variance attributed to AKL from the HGLM was 5.0%.

The QTL scan for kype height (KH) returned one locus on linkage group SSA1 (electronic supplementary material, figure S4b). The correlation between KH and SSA1 was strong among the offspring of parents A9 which displayed significant differences in KH values depending on which parental allele was inherited. The observed HK values were, respectively, 2.35 ± 1.31 and 0.66 ± 0.16 ($t = 3.8$, d.f. = 8 $p = 0.004$) for the offspring that inherited haplotype 1 and haplotype 2 from parent A9. This QTL was, however, non-significant for AKH. No genomic regions were significantly associated with AKH in our study. Estimated relative proportion of QTL variance attributed to KH from the HGLM was 6.6%.

# 4. Discussion

To our knowledge, this is the first study to investigate the potential influence of domestication on head morphology in sexually mature adult male Atlantic salmon where sexual selection has been relaxed compared to the wild. Since the early 1970s, farmed Atlantic salmon have undergone more than 12 generations of directional selection for both economically important traits and traits attributed to intrinsic selection pressures of the aquaculture environment, all in addition to the relaxation of natural selection [13,45]. We therefore hypothesized that as domesticated salmon have not been exposed to sexual selection since the founding individuals were taken into fish farms more than 12 generations ago, changes in head morphology and in particular the kype, suggested from previous studies to be linked with male reproductive success [12,37], was likely. After analysing 528 mature male salmon from multiple wild, hybrid and a domesticated strain, all of which had been reared under identical conditions from hatching onwards, we found small yet significant differences in fork length-AKH and fork length-AKL among the strains investigated. The domesticated strain displayed a reduction in fork length-AKH, both compared to the F1 hybrid strains and two of the wild strains. The reduced kype height trait identified here was not female mimicry, as highlighted by the geometric morphometric analysis, which showed clear head shape separation between female and male fish of all ages and strains. We also found a significant increase in fork length-AKL in the domesticated and hybrid MF strain when compared to the wild Arna strain, suggesting that the length of the lower jaw, or kype length here, is not an important feature in sexual selection. Collectively, our findings suggest that the relaxation of sexual selection during nearly 50 years selective breeding has driven shifts in kype structure in domesticated Atlantic salmon.

## 4.1. Domestication, sexual selection and the kype

Earlier studies have strongly indicated that the male kype is important for the reproductive success of adult male Atlantic salmon in the wild [12], although never independent of other correlative effects [9]. In salmon aquaculture, breeding programmes circumvent the potential evolutionary significance of any sexual characteristics as fish are paired after manual stripping of gametes, according to the human-determined breeding value of individual fish. Thus, we predict that the loss of female selection and male–male competition, though with the continuation of strong aquaculture trait selection and increased performance under aquaculture conditions will impact the development of sexual characteristics. Resource allocation prioritizes traits directly or inadvertently selected for, such as fast growth, delayed maturation, high survival and tempered stress response [45]. Similar trade-offs have been observed in other captively bred salmonids, with reduced kype lengths seen in hatchery-reared female Pacific salmon (*Oncorhynchus kisutch*) [14] and, to a lesser extent, ranched male sea-trout (*Salmo trutta*) [15], when compared to wild counterparts. The results of the current study support the hypothesis of reduced secondary sexual characters under artificial breeding, in concordance with the previous studies, whereby domesticated individuals exhibit significantly reduced AKH in comparison to two out of three wild strains. It also provides further evidence of the kype's importance in sexual selection, specifically kype height, and the likely energetic cost of producing a larger kype height [2,37], which has resulted in its reduction within an aquaculture setting.

The domesticated strain used here, Mowi, was founded in part by individuals from the river Vosso. Therefore, the reduction in AKH in the domesticated strain when compared with the Vosso strain was of particular interest, even though not significant ($p > 0.05$). It is noteworthy that other wild populations have also contributed to the Mowi strain, and not exclusively Vosso.

A significant increase in AKL was observed between both the domesticated and hybrid MF strains when compared to the wild Arna strain, suggesting that either AKL is not important in sexual selection, or it is indirectly selected for in the domestication process. For example, kype length could be more important in male–male competition, rather than in direct female mate choice. Male–male competition may still be present in aquaculture with individuals competing for feed [46] instead of females; therefore, kype length could still be selected for, while kype height would not. Additionally, there could also be a possible genetic linkage between traits under selection in aquaculture (e.g. fork length) and kype length. What must also be highlighted, however, is that the prediction of significantly smaller AKL in wild strains was only significant between Arna, and not Vosso or Figgjo when compared to the domesticated strain.

It is also possible that as we detected a very strong relationship between kype length and fork length, with less variance than the relationship between kype height and fork length, this could explain why the domesticated strain has significantly larger AKL than wild strains, even after adjustment for fork length; highlighting the difficulty in removing body size effect in morphological characters. Overall, it suggests that kype length, as described here, is not important in sexual selection.

## 4.2. Genetic basis of the kype

The results indicated the presence of loci that control kype length independently from fork length (a true head morphology QTL), whereas kype height is associated only with a QTL when the measure is not adjusted to fork length. Such an observation indicates that SSA1, which is associated with KH, is likely to be a QTL for body size, rather than for kype production. It is also important to consider that the power to detect QTLs is marginal in this study, and therefore, the fact that we do not detect any QTL for AKH does not mean there are no genetic factors controlling kype height. For a QTL to be detected, the following conditions have to be met: (i) the SNP genotype has to be informative within a given family so we can trace back each F1 allele to the parental alleles, and (ii) the difference in phenotype produced by the two alleles must be significant.

Despite the conflicting results between the QTL for KL and KH, adjusted or otherwise, the QTLs that were identified here demonstrate potential genomic regions for genetic control of kype variability. Further study into such QTL regions in wild–domesticated experiments would be the next step in understanding the evolutionary history of the kype.

## 4.3. Peak age in sexually mature males

Both the AKL and AKH were significantly larger in 2SW individuals, when compared to 1SW and 3SW individuals, with no significant interaction term between SW and strain. Such a trend demonstrates that the increased AKL and AKH seen in 2SW is a general feature of all strains. If a larger kype height is associated with improved reproductive success, as has been suggested in previous studies [12], this could suggest that mature males reach their maximum physical attractiveness to females in their second SW, with this then declining significantly as they enter their third SW. It could also be indicative of other life-history strategies. As males reach peak body size at the 3SW stage, it is possible that these individuals no longer need to invest as heavily in secondary sexual traits, simply due to their larger size compared to younger males; this would assume that kype size is of secondary importance, after body size, in competing for females, if the males are an order of magnitude larger than their conspecifics. Likewise, 1SW males are going to have the smallest body size, having spent less time at sea, which will also have reduced their risk of mortality at sea, perhaps corresponding to a low-risk low-investment strategy on spawning grounds; depending instead on chance matings rather than competition. Leaving 2SW males as generalists that have to invest in the kype to compete with the larger males, as well as an inability to adopt more chance matings due to their size.

## 4.4. Ecological implications and further research

With the rapid increase in Atlantic salmon in aquaculture, starting in the 1970s, there has been a greater proportion of domesticated individuals escaping into the wild. Research into the impact of these escapees

on wild populations first started in the 1990s [47] but has continued consistently since [13]. There is now strong evidence demonstrating that spawning success in domesticated individuals is lower than that of wild individuals, with domesticated males showing disproportionately lower spawning success than domesticated females [18,19,48]. The reduction in breeding success seen in domesticated males, more so than domesticated females, is due to a reduction in courtship with wild females, which has been attributed to inappropriate mating behaviour [19]. Results here indicate, however, that domesticated individuals may not only be disadvantaged due to shifts in behaviour as shown in the literature, but also by the size of their kype. Moreover, genetic control over kype height is additive, as shown by the reduction in kype height in the hybrid FM strain (Figgjo ♀ × Mowi ♂), with potential reductions in breeding success in populations with high levels of introgression. Successful spawning between wild males and farmed females during the breeding season is typically a small proportion of spawning individuals, as mentioned previously, and so hybrids such as the hybrid FM strain (Figgjo ♀ × Mowi ♂) here would be in the minority.

To elucidate the determinants and dynamics of observed reductions in fitness in domesticated individuals through disrupted mating strategies, kype measurement should be integrated into behavioural studies examining reproductive behaviour, as in Järvi [12]; while also correcting for body size in a statistically robust manner. Constructing behavioural experiments that use the natural variation in AKH, and by selecting individuals with high and low AKH, or even by manipulating kype morphology through prosthetics and three-dimensional printing, are especially potent avenues of investigation. Results here also support the argument that kype length is not important in sexual selection, and that future studies should focus on measurements such as kype height. Fully understanding natural variation in wild mature male head morphology, including kype height, would also be beneficial for our understanding, exploring more Norwegian strains, as well as strains from other locations within the natural range of Atlantic salmon. Examining more wild strains in a wild common garden, to complement the hatchery style common garden shown here, would also be valuable, as intragenerational environmental influences on morphology could also be assessed. Finally, integrating measures of sexually selected traits, such as kype height, into life-history and survival datasets would provide further insights into how sexual selection operates in Atlantic salmon. What is clear, however, is that there is very little empirical evidence on the kype's role in sexual selection, and there is a multitude of ways in which we can build on the previous literature that has tried to elucidate its role [12].

In addition to established impacts of hybridization between wild and domesticated salmon, generally resulting in reduced fitness of offspring, the ongoing high incidence of escapees globally [13] and across species [49–53] is heightening concern that genetic impacts from both pre-zygotic and post-zygotic mechanisms may lower fitness. We show here that the process of domestication is likely to play a disruptive role in sexual selection through a morphological change in secondary sexual characteristics; disruption that is also seen in the hybrids between domesticated and wild individuals. Changes in sexually selected traits, as highlighted here, could also be occurring in other finfish species that are also undergoing rapid domestication, in all corners of the globe. Such changes are particularly worrying in systems where there is limited information on life history and ecology, as long-term evolutionary trajectories, as well as wild stock viability, could be undermined through increased domesticated escapees.

Animal ethics. Those working directly with the experimental animals had undergone Norwegian Food Safety Authority (NFSA) training, as is required with experimentation involving animals that are included in the Norwegian Animal Welfare Act (2010).

Data accessibility. You can find all morphometric data used in this study in the electronic supplementary material, along with the R code used for the analyses. Data pertaining to the QTL are available from [54].

Authors' contributions. W.B.P. took photographs and measurements from the fish, applied all landmarks, conducted geometric morphometric analysis, mixed effect modelling analysis, compiled data and wrote the manuscript. M.F.S. was involved in producing the fish, taking photographs, taking measurements from the fish, constructing the mixed effect models and interpretation of the results, while also being involved in the conception and design of the work. F.B. conducted all QTL analysis. I.H.M. and L.D. produced the fish used in this study, while also taking measurements from the fish, rearing the fish and taking photographs. P.G.F., S.C., M.L. and M.I.T. were involved in the conception and design of the work. F.A. conducted genetic sexing of the fish. K.A.G. and G.C. were involved in the conception and design of the work, as well as securing the funding for the project.

Competing interests. The authors declare no competing interests.

Funding. This work was funded by the Norwegian Research Council project INTERACT (grant no. 200510), and the UK Natural Environment Research Council (NERC) Envision doctoral training programme.

Acknowledgements. We would like to thank the staff at the Norwegian Institute of Marine Research's Matre Research Station, both technical and administrative, who were involved in facilitating the experiment.

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
