## [Reviewer comments · Royal Society Open Science]

Review History

RSOS-190021.R0 (Original submission)

Review form: Reviewer 1

Is the manuscript scientifically sound in its present form?

Yes

Are the interpretations and conclusions justified by the results?

No

Is the language acceptable?

Yes

Is it clear how to access all supporting data?

Yes

Do you have any ethical concerns with this paper?

No

Have you any concerns about statistical analyses in this paper?

Yes

Recommendation?

Accept with minor revision (please list in comments)

Comments to the Author(s)

This manuscript presents a nice common garden experiment involving salmon of three sea-ages, which allows testing the effect of strain and age on a male secondary sexual trait: kype. The experimental setup is adapted, although several domesticated strains would have increased its power and biological significance. The manuscript is concise and easy to read, and the illustrations are nice. I have concerns about neither ethics nor sharing of data and analysis code. However, I have the feeling that the statistical analysis, although quite sound as it is, could be improved, to fit the goal of the manuscript more closely, and to strengthen the discussion. Especially, since the main scope (according to the title) is to test the effect of age and domestication, why not include strain type (wild, domesticated or hybrid) in the mixed model, with strain nested in it? Instead, you performed all pairwise comparisons, a few of which actually making sense for your initial question.

For AKL, two significant contrasts go in the unexpected direction (Arna < Domesticated and Arna < MF). You discuss this as being a possible artefact due to the correlation of kype length with fork length, which your adjustment method may have failed to remove. This made me look for possibly better ways to account for allometry in your data. I guess you pooled all individuals from all groups in the log-log linear regressions of fork length and kype length (height), from which you take the residuals as adjusted kype length (height). However, the factors included in the subsequent LME (strain, age...) could affect fork length, kype length (height), and the allometric coefficient between both. Wouldn't it be interesting to use the approach proposed by Nakagawa et al. (<https://doi.org/10.1186/s12915-017-0448-5>), and include fork length as a covariable in the LME? Using a model like their Eqn. 8, you would be able to account for different slopes of log(FL) on log(KL) for the different groups (strains, age...). But maybe this suggestion, in addition to the previous one, would result in an overparametrized model.

More specific comments.

- Could you tell what criteria are used to select spawners in salmon farming? Could they be selected on traits that are indirectly linked to kype length or height?

- L110, 126: the fish matured at sea and were sampled there, right? You sampled them at the end of the spawning season, but they haven't spawned. Is it possible that some of them started then to resorb their kype, as it has been documented in kelts (doi: 10.1046/j.1469-7580.2003.00239.x)?

- L131: I guess you measured fork length from the tip of the maxillary (not the mandible). Please make this explicit, so that readers understand fork length does not include kype.

L177: is there a prediction for the relationship between GSR and kype characteristics?

L270: the "1" after "haplotype" has disappeared.

L279: write "in addition" instead of "in additional".

Review form: Reviewer 2 (Sean Hayes)**Is the manuscript scientifically sound in its present form?**

Yes

Are the interpretations and conclusions justified by the results?

Yes

Is the language acceptable?

Yes

Is it clear how to access all supporting data?

Yes

Do you have any ethical concerns with this paper?

No

Have you any concerns about statistical analyses in this paper?

No

Recommendation?

Accept with minor revision (please list in comments)

Comments to the Author(s)

Perry et al- evolutionary drivers of kype size

Perry et al have conducted an elegant experiment that takes advantage of time gone by since selection of specific stocks for Atlantic salmon aquaculture. Given that modern aquaculture breeding practices likely have relaxed specific selection for secondary sexual characteristics, they test whether male kype size (height and length) have diminished over ~12 generations. The work is really quite similar in concept to female selection for male tail length in swordfish and widowbirds- but more difficult to test experimentally since unlike tails- it is harder to manipulate the size of the kype. They provide some discussion as to the function of the kype- whether it provides some mechanical value in formal male-male conflict or is purely ornamental driven by female choice remains uncertain.

For the experiment- they make an effort to compare the stocks from the same historical rivers to minimize selection that might happen from breeding in different rivers and habitat types- something that is well established in pacific salmon- particularly sockeye. I have minimal concerns with the manuscript. The results are fairly unsurprising (to my preconceived opinions at least). I suspect some reviewers would argue for large family groups to better control for strong family bias's or traits but I have little doubt results would differ. For the discussion section 'ecological implications and further research'- I agree with the literature that given a range of mate choices- kype height does seem to play a role in male selection for salmon and influences success at a behavioral level. However from a management perspective focused on recovering natural populations of Atlantic salmon- I do not see evidence that kype size is directly linked to population viability- and that, at least for this trait if a bunch of small kyped males were used to recover a population- mate choice would likely 're-evolve' increased kype heights over multiple generations. An implied point of the authors is that likely other traits that might be affecting population recovery might also be 'degrading' through artificial selection and these characteristics when interbred into wild populations could have negative effects. It is identifying these critical traits where I propose future research should be directed.

I have a few minor comments for various sections and will detail them below relative to specific line numbers in the paper.

Sean Hayes

Line #-comment

43- I suspect this might be a US English for GB English issue- but should manifest have ED on the end?

49- similarly- varied to- variation?

115- please provide additional details to methods- a) the reader presumes that for the wild breedings in this experiment- the wild fish were not allowed to 'mate select' themselves, but rather were subjected to same eggs and sperm in a cup and selected by humans? B) how was selection of wild fish controlled for to prevent any bias such for particularly large kypes either in the field or by human choice in the hatchery?

209-212- I follow your logic in assigning these fish to 'female' but it would seem that if they are expressing milt- that is a primary sex characteristic- not a secondary one- and as such one might be inclined to identify them as males instead? Given it was equal across all three groups I doubt the assignment to either sex effects the results.

253-255- are you basically arguing that kype length is not relevant to the differences in male and female head morphology?

279- change additional to addition?

314-315- I suspect you are correct but could a third potential (and less likely) explanation be that insufficient variation of this trait existed in the original brood stock for relaxed selection to show a measureable change over only 12 generations?

315-316- again- I am inclined to agree with your argument- but can you fully rule out that kype length and forklength correlation is not a function of some form of genetic linkage?

345-347- would another way of saying this be that kype height is of secondary importance in female mate choice compared to total body size?

361-363- Im a little unsure the authors mean by 'but also the size of their kype'. I only see evidence in the literature for kype size just having behavioral shifts in selection preference?

366-367- So.. wild females really select for total male size potentially as a highest priority?

Review form: Reviewer 3

Is the manuscript scientifically sound in its present form?

Yes

Are the interpretations and conclusions justified by the results?

Yes

Is the language acceptable?

Yes

Is it clear how to access all supporting data?

Yes

Do you have any ethical concerns with this paper?

No

Have you any concerns about statistical analyses in this paper?

No

Recommendation?

Major revision is needed (please make suggestions in comments)

Comments to the Author(s)

Comments to authors are attached (Appendix A).

Decision letter (RSOS-190021.R0)

01-Mar-2019

Dear Mr Perry

On behalf of the Editors, I am pleased to inform you that your Manuscript RSOS-190021 entitled "Evolutionary drivers of kype size in Atlantic salmon (*Salmo salar*): domestication, age and genetics" has been accepted for publication in Royal Society Open Science subject to minor revision in accordance with the referee suggestions. Please find the referees' comments at the end of this email.

The reviewers and handling editors have recommended publication, but also suggest some minor revisions to your manuscript. Therefore, I invite you to respond to the comments and revise your manuscript.

- Ethics statement

- Data accessibility

<http://datadryad.org/submit?journalID=RSOS&manu=RSOS-190021>

- Competing interests

- Authors' contributions

- Acknowledgements

- Funding statement

Because the schedule for publication is very tight, it is a condition of publication that you submit the revised version of your manuscript before 10-Mar-2019. Please note that the revision deadline will expire at 00.00am on this date. If you do not think you will be able to meet this date please let me know immediately.

- 1) A text file of the manuscript (tex, txt, rtf, docx or doc), references, tables (including captions) and figure captions. Do not upload a PDF as your "Main Document";
- 2) A separate electronic file of each figure (EPS or print-quality PDF preferred (either format should be produced directly from original creation package), or original software format);
- 3) Included a 100 word media summary of your paper when requested at submission. Please ensure you have entered correct contact details (email, institution and telephone) in your user account;
- 4) Included the raw data to support the claims made in your paper. You can either include your data as electronic supplementary material or upload to a repository and include the relevant doi

within your manuscript. Make sure it is clear in your data accessibility statement how the data can be accessed;

5) All supplementary materials accompanying an accepted article will be treated as in their final form. Note that the Royal Society will neither edit nor typeset supplementary material and it will be hosted as provided. Please ensure that the supplementary material includes the paper details where possible (authors, article title, journal name).

on behalf of Prof Kevin Padian (Subject Editor)
openscience@royalsociety.org

Reviewer comments to Author:

Reviewer: 1

Comments to the Author(s)

This manuscript presents a nice common garden experiment involving salmon of three sea-ages, which allows testing the effect of strain and age on a male secondary sexual trait: kype. The experimental setup is adapted, although several domesticated strains would have increased its power and biological significance. The manuscript is concise and easy to read, and the illustrations are nice. I have concerns about neither ethics nor sharing of data and analysis code. However, I have the feeling that the statistical analysis, although quite sound as it is, could be improved, to fit the goal of the manuscript more closely, and to strengthen the discussion.

Especially, since the main scope (according to the title) is to test the effect of age and domestication, why not include strain type (wild, domesticated or hybrid) in the mixed model, with strain nested in it? Instead, you performed all pairwise comparisons, a few of which actually making sense for your initial question.

For AKL, two significant contrasts go in the unexpected direction (Arna<Domesticated and Arna<MF). You discuss this as being a possible artefact due to the correlation of kype length with fork length, which your adjustment method may have failed to remove. This made me look for possibly better ways to account for allometry in your data. I guess you pooled all individuals from all groups in the log-log linear regressions of fork length and kype length (height), from which you take the residuals as adjusted kype length (height). However, the factors included in the subsequent LME (strain, age...) could affect fork length, kype length(height), and the allometric coefficient between both. Wouldn't it be interesting to use the approach proposed by Nakagawa et al. (<https://doi.org/10.1186/s12915-017-0448-5>), and include fork length as a covariable in the LME? Using a model like their Eqn. 8, you would be able to account for different slopes of log(FL) on log(KL) for the different groups (strains, age...). But maybe this suggestion, in addition to the previous one, would result in an overparametrized model.

More specific comments.

- Could you tell what criteria are used to select spawners in salmon farming? Could they be selected on traits that are indirectly linked to kype length or height?

- L110, 126: the fish matured at sea and were sampled there, right? You sampled them at the end of the spawning season, but they haven't spawned. Is it possible that some of them started then to resorb their kype, as it has been documented in kelts (doi: 10.1046/j.1469-7580.2003.00239.x)?

- L131: I guess you measured fork length from the tip of the maxillary (not the mandible). Please make this explicit, so that readers understand fork length does not include kype.

L177: is there a prediction for the relationship between GSR and kype characteristics?

L270: the "1" after "haplotype" has disappeared.

L279: write "in addition" instead of "in additional".

Reviewer: 2

Comments to the Author(s)

Perry et al- evolutionary drivers of kype size

Perry et al have conducted an elegant experiment that takes advantage of time gone by since selection of specific stocks for Atlantic salmon aquaculture. Given that modern aquaculture breeding practices likely have relaxed specific selection for secondary sexual characteristics, they test whether male kype size (height and length) have diminished over ~12 generations. The work is really quite similar in concept to female selection for male tail length in swordfish and widowbirds- but more difficult to test experimentally since unlike tails- it is harder to manipulate the size of the kype. They provide some discussion as to the function of the kype- whether it provides some mechanical value in formal male-male conflict or is purely ornamental driven by female choice remains uncertain.

For the experiment- they make an effort to compare the stocks from the same historical rivers to minimize selection that might happen from breeding in different rivers and habitat types- something that is well established in pacific salmon- particularly sockeye. I have minimal concerns with the manuscript. The results are fairly unsurprising (to my preconceived opinions at least). I suspect some reviewers would argue for large family groups to better control for strong family bias's or traits but I have little doubt results would differ. For the discussion section 'ecological implications and further research'- I agree with the literature that given a range of mate choices- kype height does seem to play a role in male selection for salmon and influences success at a behavioral level. However from a management perspective focused on recovering natural populations of Atlantic salmon- I do not see evidence that kype size is directly

linked to population viability- and that, at least for this trait if a bunch of small kyped males were used to recover a population- mate choice would likely 're-evolve' increased kype heights over multiple generations. An implied point of the authors is that likely other traits that might be affecting population recovery might also be 'degrading' through artificial selection and these characteristics when interbred into wild populations could have negative effects. It is identifying these critical traits where I propose future research should be directed.

I have a few minor comments for various sections and will detail them below relative to specific line numbers in the paper.

Sean Hayes

Line #-comment

43- I suspect this might be a US English for GB English issue- but should manifest have ED on the end?

49- similarly- varied to- variation?

115- please provide additional details to methods- a) the reader presumes that for the wild breedings in this experiment- the wild fish were not allowed to 'mate select' themselves, but rather were subjected to same eggs and sperm in a cup and selected by humans? B) how was selection of wild fish controlled for to prevent any bias such for particularly large kypes either in the field or by human choice in the hatchery?

209-212- I follow your logic in assigning these fish to 'female' but it would seem that if they are expressing milt- that is a primary sex characteristic- not a secondary one- and as such one might be inclined to identify them as males instead? Given it was equal across all three groups I doubt the assignment to either sex effects the results.

253-255- are you basically arguing that kype length is not relevant to the differences in male and female head morphology?

279- change additional to addition?

314-315- I suspect you are correct but could a third potential (and less likely) explanation be that insufficient variation of this trait existed in the original brood stock for relaxed selection to show a measureable change over only 12 generations?

315-316- again- I am inclined to agree with your argument- but can you fully rule out that kype length and forklength correlation is not a function of some form of genetic linkage?

345-347- would another way of saying this be that kype height is of secondary importance in female mate choice compared to total body size?

361-363- Im a little unsure the authors mean by 'but also the size of their kype'. I only see evidence in the literature for kype size just having behavioral shifts in selection preference?

366-367- So.. wild females really select for total male size potentially as a highest priority?

Reviewer: 3

Comments to the Author(s)

Comments to authors are attached

Author's Response to Decision Letter for (RSOS-190021.R0)

See Appendix B.

Decision letter (RSOS-190021.R1)

20-Mar-2019

Dear Mr Perry,

I am pleased to inform you that your manuscript entitled "Evolutionary drivers of kype size in Atlantic salmon (*Salmo salar*): domestication, age and genetics" is now accepted for publication in Royal Society Open Science.

on behalf of Mr Andrew Dunn (Associate Editor) and Kevin Padian (Subject Editor)
openscience@royalsociety.org

Associate Editor Comments to Author (Mr Andrew Dunn):
Associate Editor: 1
Comments to the Author:
(There are no comments.)

Reviewer comments to Author:

Appendix A

RSOS-190021 – Evolutionary drivers of kype size in Atlantic salmon (*Salmo salar*): domestication, age and genetics

Comments to authors

This paper investigates changes in a secondary sex characteristic (kype) associated with domestication in Atlantic salmon. The study compares kype traits (length and height) in wild, domestic, and hybrid strains of salmon reared under common garden conditions. The study finds differences between strains and age-at-maturity in kype traits. Specifically, the study finds that adjusted kype height is significantly lower in domestic salmon, suggesting a role for relaxed sexual selection associated with domestication. The study identifies some loci that are possibly associated with kype traits. In addition, the study also compares head morphology using geometric morphometrics, and generally finds differences between sexes and ages that are not associated with domestication. This last point (Geometric morphometrics analysis) is not addressed in the Discussion and warrants at least a sentence or two. The study should also better define the purpose/objective of this analysis in the paper.

Overall the paper reads well and addresses some important issues relating to domestication. My primary concern is the difference found between adjusted kype height (AKH) and adjusted kype length (AKL). AKH is significantly lower in domestic fish relative to wild, supporting the conclusion “that the relaxation of sexual-selection during nearly fifty years selective breeding has driven shifts in kype structure in domesticated Atlantic salmon” (Lines 288-289). However, the paper also finds that AKL is different between strains ($p < 0.01$), where domestic fish have the greater AKL (albeit posthoc pairwise comparisons were not significant). This is not clearly explained in the same part of the discussion (Line 288-289) or in the Abstract (Line 30). The authors suggest that AKL is “is not important in sexual selection, or it is indirectly selected for in the domestication process” (Line 314-315). The authors also suggest there is little genetic variation for AKL trait because kype length is highly correlated with body length (more so than KH); but R^2 for both traits with body length is very high ($R^2 = 0.87$ and 0.81). However, the authors find the “presence of loci that control kype length independently from fork length” (Line 325-326). This seems to suggest that there is genetic variation associated with kype length, and more so than AKH (where

no QTLs were found). I think this part of the paper needs some revising to provide better interpretation of the results. Is it also possible that this trait AKL is not really representative of the kype but rather just the length of the lower jaw, regardless of whether the individual has a kype or not. This is an important consideration that should be addressed. This would allow future mate choice or other studies to focus on the important trait.

In addition, would it be possible to score individuals in a more binary way for presence or absence of a kype? Or potentially a “kype index” ranging from 0 (no kype) to 5 (full kype) and allowing for intermediates? From the photographs provided, it seems that something like that may be possible and it would be interesting to explore this relationship, particularly for the genetic analyses.

Aside from these concerns, my only other major comment is that more information is needed for the QTL results. A Manhattan plot or table would be useful. If other studies examine QTLs for kype traits in the future, then it is important to know where these QTLs are located for comparison. Further, it would be helpful to know how much variation can be explained by the QTL.

Specific comments:

Abstract

Lines 22-25 – This sentence should be revised. Not clear.

Line 30 – Worth mentioning that the same relationship was not found for AKL, in fact, AKL was greater in domestic strain compared to the wild strain.

Introduction - Overall the introduction provides great context for the study and reads well.

Line 52 – include (i.e., nests) for non-salmonid readers.

Lines 96-100 – No mention of genetic analyses in this section?

Methods - The methods are generally clear and analyses outlined appear to be appropriate and statically sound.

Line 109 – If strains are mixed at the eyed-egg stage, how is their origin known?

Line 126 – Was this the first time that each male matured? Could they have spawned previously provided that Atlantic salmon are iteroparous?

Line 140 – What was the origin of the females? What were their ages?

Line 196-198 – There are many factors in the model. Could the model be over correcting (accounting for) for real variation associated with the trait, thus reducing the power to detect true QTL?

Line 227 – Authors should make it clear that AKL is actually larger in domestic strain. Reword to indicate that AKL was greater in domesticated compared to Arna (similar to wording indicated in the AKH comparison; Line 238)

Line 255 – Is this worded correctly? Should it say “does not constitute female head morphology”?

Line 257 – What is causing the separation between 1/2SW and 3SW fish?

Lines 261-272 – Please include figure (Manhattan plot) for these QTL analyses. Or at least a table of genomic position of these SNPs. How much variation can be explained by genotype?

Line 270 – Missing haplotype “1” here?

Line 288 – It may be appropriate to indicate that adjusted kype length was actually larger in domestic strain here. Or to provide some information about the lack of difference in AKL.

Line 305-306 - This sentence is a bit misleading. Perhaps it would be more appropriate to end this section (at the end of these few paragraphs) with this sentence/thought, and to indicate specifically that it is AKH (not AKL) that is important for sexual selection.

Line 311-312 – This is a bit confusing, as AKL was significantly different between strains ($p < 0.01$) (Domestic and Arna, $p = 0.02$)(Arna and hybrid MF, $p = 0.03$)? I think the posthoc pairwise comparisons were not significant, but if that is true, these p -values associated with pairwise comparisons are not useful. Please clarify.

Line 314-315 – Why might these traits be under different selective pressure? Are there potential reasons why kype height may be more important for sexual selection? I think this would be interesting to consider.

Line 325 – Maybe I have missed something, but the authors argue that there is little genetic variation for kype length (line 316-317), but then show that there are loci that control kype length independent of fork length (Line 325-326). This is confusing and this part of the discussion should be revised. It would also be useful as indicated in the Results to know how much variation can be explained by these loci.

Line 339-352 – It would be useful here to review the evidence on kype characteristics and sexual selection. What has been found previously?

Appendix B

Reviewer 1:

Comments to the Author(s)

This manuscript presents a nice common garden experiment involving salmon of three sea-ages, which allows testing the effect of strain and age on a male secondary sexual trait: kype. The experimental setup is adapted, although several domesticated strains would have increased its power and biological significance. The manuscript is concise and easy to read, and the illustrations are nice. I have concerns about neither ethics nor sharing of data and analysis code.

However, I have the feeling that the statistical analysis, although quite sound as it is, could be improved, to fit the goal of the manuscript more closely, and to strengthen the discussion.

Especially, since the main scope (according to the title) is to test the effect of age and domestication, why not include strain type (wild, domesticated or hybrid) in the mixed model, with strain nested in it? Instead, you performed all pairwise comparisons, a few of which actually making sense for your initial question.

For AKL, two significant contrasts go in the unexpected direction (Arna < Domesticated and Arna < MF). You discuss this as being a possible artefact due to the correlation of kype length with fork length, which your adjustment method may have failed to remove. This made me look for possibly better ways to account for allometry in your data. I guess you pooled all individuals from all groups in the log-log linear regressions of fork length and kype length (height), from which you take the residuals as adjusted kype length (height). However, the factors included in the subsequent LME (strain, age...) could affect fork length, kype length(height), and the allometric coefficient between both. Wouldn't it be interesting to use the approach proposed by Nakagawa et al. (<https://doi.org/10.1186/s12915-017-0448-5>), and include fork length as a covariable in the LME? Using a model like their Eqn. 8, you would be able to account for different slopes of log(FL) on log(KL) for the different groups (strains, age...). But maybe this suggestion, in addition to the previous one, would result in an overparametrized model.

Response: Thank you very much for your insightful comments, and constructive feedback. Please find my response to your main points:

- 1. Using type (wild, farmed, hybrid) with strain nested – This is a very good point, and it is something that I had included in the first models, and unsurprisingly, it**

showed that there was a difference between the types. However, after some discussion, we decided that it would be more honest to simply show strain, as not to whitewash over the ‘non-significant to $P < 0.05$ ’ difference between the domesticated strain and Vosso.

2. Controlling for allometry was probably one of the biggest challenges we had in the analysis, with us finally settling (like you describe) with a log regression between the kype length and fork length among all individuals, avoiding divisions due to the statistical problems highlighted by Nakagawa et al. As for including fork length as a covariate in the model, I think you are also right about the overparameterization. I have consulted the authors Alain F. Zuur and Elena N. Ieno of the book ‘Mixed Effects Models and Extensions in Ecology with R’ (DOI: 10.1007/978-0-387-87458-6). They too suggested that feeding an appropriate size adjusted metric for kype length, like residuals from a regression would be the best way forward, as mixed effect models do not deal well with strongly correlated covariates. The paper by Nakagawa et al. is very informative, however, and I will certainly bear this in mind for future work.

More specific comments.

- Could you tell what criteria are used to select spawners in salmon farming? Could they be selected on traits that are indirectly linked to kype length or height?

Response: Breeding programs will select for things such as body size, delayed maturation, high survival, and tempered stress response. I would say that there might be an indirect link between kype length and body size, but I think if we were to see that there was an indirect link here with kype height, we would not be seeing a reduction in the domesticated individuals.

- L110, 126: the fish matured at sea and were sampled there, right? You sampled them at the end of the spawning season, but they haven't spawned. Is it possible that some of them started then to resorb their kype, as it has been documented in kelts (doi: 10.1046/j.1469-7580.2003.00239.x)?

Response: Although there are observations of kype resorption of the kype in Atlantic salmon, the extent of this is unknown. The study by Witten et al. (2003) you refer to summarises that from their work: “our present observations neither support suggestions of periodic return of the lower jaw to its previous shape”. Even though

there may be evidence for demineralisation, this may not equate to a change in size. As we looked at the kype in a common garden setting, resorption, if it were to happen, should have occurred at the same time across strains. The males from different sea winters were also terminated at roughly the same time each year (10 days apart, max). I should also imagine that there are a variety of cues that would trigger reabsorption other than time of year, such as a move to freshwater, which was not present in the conditions these fish were reared in.

- L131: I guess you measured fork length from the tip of the maxillary (not the mandible). Please make this explicit, so that readers understand fork length does not include kype.

Response: Addition of “most anterior point of the head to the of the middle caudal fin rays”

L177: is there a prediction for the relationship between GSR and kype characteristics?

Response: GSR was not a significant term in the model. We did see a positive relationship between AKH and gonadosomatic index (GSI - $GSI = [\text{gonad weight} / \text{total tissue weight}] \times 100$), but this was likely due to the fact that allometry was not properly adjusted for by this metric.

L270: the "1" after "haplotype" has disappeared.

Response: Fixed

L279: write "in addition" instead of "in additional".

Response: Fixed

Reviewer: 2

Comments to the Author(s)

Perry et al- evolutionary drivers of kype size

Perry et al have conducted an elegant experiment that takes advantage of time gone by since selection of specific stocks for Atlantic salmon aquaculture. Given that modern aquaculture breeding practices likely have relaxed specific selection for secondary sexual characteristics, they test whether male kype size (height and length) have diminished over ~12 generations. The work is really quite similar in concept to female selection for male tail length in

swordfish and widowbirds- but more difficult to test experimentally since unlike tails- it is harder to manipulate the size of the kype. They provide some discussion as to the function of the kype- whether it provides some mechanical value in formal male-male conflict or is purely ornamental driven by female choice remains uncertain.

For the experiment- they make an effort to compare the stocks from the same historical rivers to minimize selection that might happen from breeding in different rivers and habitat types- something that is well established in pacific salmon- particularly sockeye. I have minimal concerns with the manuscript. The results are fairly unsurprising (to my preconceived opinions at least). I suspect some reviewers would argue for large family groups to better control for strong family bias's or traits but I have little doubt results would differ. For the discussion section 'ecological implications and further research'- I agree with the literature that given a range of mate choices- kype height does seem to play a role in male selection for salmon and influences success at a behavioral level. However from a management perspective focused on recovering natural populations of Atlantic salmon- I do not see evidence that kype size is directly linked to population viability- and that, at least for this trait if a bunch of small kyped males were used to recover a population- mate choice would likely 're-evolve' increased kype heights over multiple generations. An implied point of the authors is that likely other traits that might be affecting population recovery might also be 'degrading' through artificial selection and these characteristics when interbred into wild populations could have negative effects. It is identifying these critical traits where I propose future research should be directed.

I have a few minor comments for various sections and will detail them below relative to specific line numbers in the paper.

Response: Thank you for your kind comments and constructive feedback. Please find my response to your main points:

I would agree that kype size is not going to be the key determinant of the viability of a river. I also agree that if a haplotype for a small kype were to spread through the population, if all of that population then had small kypes, mate choice would just 're-evolve'.

I do think, however, that the kype could be part of why we see farmed escapees being less competitive, and because we see an intermediate kype phenotype in hybrids, this could also be related to their fitness. As salmon populations are highly structured

genetically and physically, it would not be un-thinkable that unfavourable haplotypes, such as those for smaller kype size, could build up in areas of high escapees, perhaps causing isolation from larger populations, and contributing to a spiral of decline (associated with a myriad of other typically introgressed phenotypes, and human pressures).

Line #-comment

43- I suspect this might be a US English for GB English issue- but should manifest have ED on the end?

Response: Fixed

49- similarly- varied to- variation?

Response: Fixed

115- please provide additional details to methods- a) the reader presumes that for the wild breedings in this experiment- the wild fish were not allowed to ‘mate select’ themselves, but rather were subjected to same eggs and sperm in a cup and selected by humans? B) how was selection of wild fish controlled for to prevent any bias such for particularly large kypes either in the field or by human choice in the hatchery?

Response to A: Addition of “Artificial fertilisation of gametes took place on arrival at the IMR Matre Research station”

Response to B: Wild parents were caught by angling (Figgjo, Vosso), nets (Vosso) or salmon trap. Farmed parents were chosen at ‘random’ from the breeding station at Askøy. This information is included in “Solberg et al. (2014) (information on the cohort produced in 2012)”. I have tried to highlight what information can be found in Solberg et al. (2014) a bit better in the methods.

209-212- I follow your logic in assigning these fish to ‘female’ but it would seem that if they are expressing milt- that is a primary sex characteristic- not a secondary one- and as such one might be inclined to identify them as males instead? Given it was equal across all three groups I doubt the assignment to either sex effects the results.

Response: I did run the analysis both with and without these individuals, and like you say, it made no difference.

253-255- are you basically arguing that kype length is not relevant to the differences in male and female head morphology?

Response: Apologies – should read does NOT constitute!

279- change additional to addition?

Response: Fixed

314-315- I suspect you are correct but could a third potential (and less likely) explanation be that insufficient variation of this trait existed in the original brood stock for relaxed selection to show a measureable change over only 12 generations?

Response: Fish from many rivers were used in the origin of the Mowi strain, so I should think that there would have been variation in this trait.

315-316- again- I am inclined to agree with your argument- but can you fully rule out that kype length and forklength correlation is not a function of some form of genetic linkage?

Response: Addition of “Additionally, there could also be possible genetic linkage between traits under selection (e.g. fork length) and kype length”.

345-347- would another way of saying this be that kype height is of secondary importance in female mate choice compared to total body size?

Response: Yes, see addition of “; this would assume that kype size is of secondary importance, after body size, in competing for females, if the males are an order of magnitude larger than their conspecifics.”

361-363- Im a little unsure the authors mean by ‘but also the size of their kype’. I only see evidence in the literature for kype size just having behavioral shifts in selection preference?

Response: Sentence clarified

366-367- So.. wild females really select for total male size potentially as a highest priority?

Response: Quite possibly. Jarvi (1990) did show that males with a larger body size also increased in dominance rank. One experimental problem that I think Javi had was separating the effects of body size and kype size, due to allometry – this is why I think experiments in manipulating kype size might be of interest in getting to the bottom of which is more important. Like you say in your comments about swordfish and widowbirds studies, however, trying to manipulate the kype would be a bit of a battle!

Reviewer 3:

This paper investigates changes in a secondary sex characteristic (kype) associated with domestication in Atlantic salmon. The study compares kype traits (length and height) in wild, domestic, and hybrid strains of salmon reared under common garden conditions.

The study finds differences between strains and age-at-maturity in kype traits.

Specifically, the study finds that adjusted kype height is significantly lower in domestic salmon, suggesting a role for relaxed sexual selection associated with domestication. The study identifies some loci that are possibly associated with kype traits. In addition, the study also compares head morphology using geometric morphometrics, and generally finds differences between sexes and ages that are not associated with domestication. This last point (Geometric morphometrics analysis) is not addressed in the Discussion and warrants at least a sentence or two. The study should also better define the purpose/objective of this analysis in the paper.

Overall the paper reads well and addresses some important issues relating to domestication. My primary concern is the difference found between adjusted kype height (AKH) and adjusted kype length (AKL). AKH is significantly lower in domestic fish relative to wild, supporting the conclusion “that the relaxation of sexual-selection during nearly fifty years selective breeding has driven shifts in kype structure in domesticated Atlantic salmon” (Lines 288-289). However, the paper also finds that AKL is different between strains ($p < 0.01$), where domestic fish have the greater AKL (albeit posthoc pairwise comparisons were not significant). This is not clearly explained in the same part of the discussion (Line 288-289) or in the Abstract (Line 30). The authors suggest that AKL is “is not important in sexual selection, or it is indirectly selected for in the domestication process” (Line 314-315). The authors also suggest there is little genetic variation for AKL trait because kype length is highly correlated with body length (more so than KH); but R^2 for both traits with body length is very high ($R^2 = 0.87$ and 0.81). However, the authors find the “presence of loci that control kype length independently from fork length” (Line 325-326). This seems to suggest that there is genetic variation associated with kype length, and more so than AKH (where no QTLs were found). I think this part of the paper needs some revising to provide better interpretation of the results. Is it also possible that this trait AKL is not really representative of the kype but rather just the length of the lower jaw, regardless of whether the individual has a kype or not. This is an important consideration that should be addressed. This would allow future mate choice or other studies to focus on the important trait.

In addition, would it be possible to score individuals in a more binary way for presence or absence of a kype? Or potentially a “kype index” ranging from 0 (no kype) to 5 (full kype) and allowing for intermediates? From the photographs provided, it seems that something like that may be possible and it would be interesting to explore this relationship, particularly for the genetic analyses.

Aside from these concerns, my only other major comment is that more information is needed for the QTL results. A Manhattan plot or table would be useful. If other studies examine QTLs for kype traits in the future, then it is important to know where these QTLs are located for comparison. Further, it would be helpful to know how much variation can be explained by the QTL.

Response:

Thank you very much for your comprehensive engagement with all aspects of the paper, your clear suggestions and constructive feedback. Please find my responses to the themes you showed concern over:

- 1. No mention of geometric morphometrics in the discussion: This is a good point, addition of “The reduced kype height trait identified here was not female mimicry, as highlighted by the geometric morphometric analysis, which showed clear head shape separation between female and male fish of all ages and strains”**
- 2. Scoring individuals on kype size: We did try to sort individuals into a kype index (0-5) like you suggest, and even into groups such as ‘kype’ and ‘no kype’. However, after looking through a number of individuals, it was clear that this was quite an ambiguous task, because of the continuum of sizes you encounter. We decided not to try and plough on ahead doing this for all 528 fish, and instead focused on kype height and kype length.**
- 3. Increased AKL in domesticated individuals: I agree that the results surrounding AKL are a bit confused, and I think it reflects the confusion we had when first interpreting the results. I agree that both kype height and kype length are highly correlated with fork length. I also agree that the results from the QTL are also the opposite of what you might find with the argument we made regarding little genetic variation for kype length. I have reconstructed this section of the discussion, further talking about biological processes that may have caused the results we see, but also including the methodological artefacts that could have also contributed (removing the little genetic variation comments). Finally ending,**

in what I think is the real take home message, that kype length is not important in sexual selection. I have emphasised the results of the AKL in the abstract. I have also added that future studies should focus on kype height rather than kype length in the ‘future research’ section. I hope this makes everything a bit clearer.

- 4. More information needed on QTL: Manhattan plot included in the supplementary materials, and variation explained by QTL for AKH and AKL was added to the results.**

Specific comments:

Abstract

Lines 22-25 – This sentence should be revised. Not clear.

Response: Fixed

Line 30 – Worth mentioning that the same relationship was not found for AKL, in fact, AKL was greater in domestic strain compared to the wild strain.

Response: Addition of “Fork length-adjusted kype length showed an increase in domesticated individuals, highlighting that kype length, as it is defined here, may not be an important in cue mate choice.”

Line 52 – include (i.e., nests) for non-salmonid readers.

Response: Fixed

Lines 96-100 – No mention of genetic analyses in this section?

Response: Addition of “alongside a quantitative trait loci analysis”

Line 109 – If strains are mixed at the eyed-egg stage, how is their origin known?

Response: A fin clip is taken from the fish and a pit tag is inserted, the PIT tag is then scanned when the fish has been terminated, allowing us to match the fish with the results from the parentage analysis performed on DNA from the finclip: “Procedures included DNA identification and subsequent PIT tagging of all individuals for identification.”

Line 126 – Was this the first time that each male matured? Could they have spawned previously provided that Atlantic salmon are iteroparous?

Response:

Line 140 – What was the origin of the females? What were their ages?

Response: Please see table 1

Line 196-198 – There are many factors in the model. Could the model be over correcting

(accounting for) for real variation associated with the trait, thus reducing the power to detect true QTL?

Response: The covariates included in the model only account for about 12% of the phenotype variation (Adjusted kype length) which means the 88% of the variation is still available in the model to fit QTL effects. The covariates in the model do not seem to be an obstacle to the detection of QTL due to soaking the available phenotypic variance.

Line 227 – Authors should make it clear that AKL is actually larger in domestic strain. Reword to indicate that AKL was greater in domesticated compared to Arna (similar to wording indicated in the AKH comparison; Line 238)

Response: Addition of “The difference in AKL among strains was driven by the significantly larger AKL in the domesticated strain (estimated mean = 0.012) when compared to Arna (estimated mean = - 0.022)”

Line 255 – Is this worded correctly? Should it say “does not constitute female head morphology”?

Response: Fixed

Line 257 – What is causing the separation between 1/2SW and 3SW fish?

Response: Addition of “The differences in male head shape between sea winters was summarised along PC1 (Proportion of variance = 46.5 %) and was characterised by a dorsal shift of features such as the eye, posterior point of the gill plate and the maxillary bone (supplementary figure 3).” And addition analysis which is contained in supplementary materials.

Lines 261-272 – Please include figure (Manhattan plot) for these QTL analyses. Or at least a table of genomic position of these SNPs. How much variation can be explained by genotype?

Response: Please see the addition of QTL plot in the supplementary materials, and references within the QTL results. Also, the addition of “Estimated relative proportion of QTL variance attributed to AKL from the Hierarchical Generalized Linear Model (HGLM) was 5.0 %.” and “Estimated relative proportion of QTL variance attributed to KH from the HGLM was 6.6 %.”

Line 270 – Missing haplotype “1” here?

Response: Fixed

Line 288 – It may be appropriate to indicate that adjusted kype length was actually larger in domestic strain here. Or to provide some information about the lack of difference in AKL.

Response: Addition of “We also found a significant increase in fork-length adjusted kype length in the domesticated and hybrid MF strain when compared to the wild Arna strain, suggesting that the length of the lower jaw, or kype length here, is not an important feature in sexual selection.”

Line 305-306 - This sentence is a bit misleading. Perhaps it would be more appropriate to end this section (at the end of these few paragraphs) with this sentence/thought, and to indicate specifically that it is AKH (not AKL) that is important for sexual selection.

Response: Sentence clarified “It also provides further evidence of the kype’s importance in sexual selection, specifically kype height, and the likely energetic cost of producing a larger kype heights [2,37], which has resulted in its reduction within an aquaculture setting”

Line 311-312 – This is a bit confusing, as AKL was significantly different between strains ($p < 0.01$) (Domestic and Arna, $p = 0.02$)(Arna and hybrid MF, $p = 0.03$)? I think the posthoc pairwise comparisons were not significant, but if that is true, these p -values associated with pairwise comparisons are not useful. Please clarify.

Response: Amended, confused sentence removed.

Line 314-315 – Why might these traits be under different selective pressure? Are there potential reasons why kype height may be more important for sexual selection? I think this would be interesting to consider.

Response: Addition of “For example, kype length could be more important in male-male competition, rather than in direct female mate choice. Male-male competition may still be present in aquaculture with individuals competing for feed [46] instead of females; therefore, kype length could still be selected for, while kype height would not. Additionally, there could also be possible genetic linkage between traits under selection (e.g. fork length) and kype length. What must also be highlighted, however, is that the prediction of significantly smaller AKL in wild strains was only seen in Arna, and not Vosso or Figgjo when compared to the domesticated strain.”

Line 325 – Maybe I have missed something, but the authors argue that there is little genetic variation for kype length (line 316-317), but then show that there are loci that control kype length independent of fork length (Line 325-326). This is confusing and this part of the discussion should be revised. It would also be useful as indicated in the

Results to know how much variation can be explained by these loci.

Response: The correlation between genotype and phenotype is only significant in one or two families which is where the QTL signal comes from.

Line 339-352 – It would be useful here to review the evidence on kype characteristics and sexual selection. What has been found previously?

Response: Addition of “ Finally, integrating measures of sexually selected traits, such as the kype, into life-history and survival datasets would provide further insights into how sexual selection operates in Atlantic salmon. What is clear, however, is that there is very little empirical evidence regarding the kype’s role in sexual selection, and there are a multitude of ways in which we can build on the previous literature that has tried to elucidate its role [12].” – there is not much out there on Atlantic salmon kypes past that 1990 study conducted by Javi. The other studies on the kype focus on its structure.